# C-SuFEx linkage of sulfonimidoyl fluorides and organotrifluoroborates

Suqin Zhao[1,4], Daming Zeng[1,4], Ming Wang ®[1] ✉ & Xuefeng Jiang ®[1,2,3] ✉

Sulfur(VI) fluoride exchange, a new type of linkage reaction, has excellent potential for application in functional molecule linkage to prepare pharmaceuticals, biomolecules, and polymers. Herein, a C-SuFEx reaction is established to achieve fast (in minutes) linkage between sulfonimidoyl fluorides and aryl/alkyl organotrifluoroborates. Potassium organotrifluoroborates are instantaneously activated via a substoichiometric amount of trimethylsilyl triflate to afford organodifluoroboranes, releasing $BF_3$ as an activating reagent in situ. This sulfur(VI) fluoride exchange technique is capable of forming S(VI)-C(alkyl), S(VI)-C(alkenyl) and S(VI)-C(aryl) bonds, demonstrating its broad scope. Natural products and pharmaceuticals with sensitive functional groups, such as valdecoxib, celecoxib and diacetonefructose, are compatible with this protocol, allowing the formation of diverse sulfoximines.

Since the concept of sulfur(VI) fluoride exchange (SuFEx) was first developed by Sharpless and coworkers in 2014[1], this efficient metal-free reaction has been widely applied in the formation of molecular linkages[2–6] in the fields of pharmaceuticals[7,8], chemical biology[9], and materials chemistry[10–12]. Sulfonimidoyl fluoride is a nitrogen isostere of sulfonyl fluoride possessing an adjustable handle for reactivity tuning and functional enhancement (Fig. 1A)[13–15]. Sulfonimidoyl fluorides have been applied to form S(VI)-O[11,12,16] and S(VI)-N[7,17,18] linkages with phenols and amines as coupling partners, aiming to generate molecular connections for rapid and efficient target-based functional discovery (Fig. 1B)[19–22]. However, the C-SuFEx [SuFEx for S(VI)-C bond formation] of sulfonimidoyl fluorides is limited to coupling with TMSCF₃[23,24] and highly active organometallic reagents such as Grignard and lithium reagents[17,25]. It is imperative to establish a C-SuFEx technique that can be applied with a stable, mild and compatible partner. Effort has been made in our group to apply the C-SuFEx technique to alkynes via an intramolecular hydride transfer process[26] and to unactivated alkenes via an intermolecular hydride transfer process[27], which requires a stoichiometric activating reagent for S(VI)-C bond construction[28]. Potassium organotrifluoroborates possess exceptional stabilities toward air and moisture[29–35], and the electronegativity of potassium (0.8), which differs from those of fluorine (4.0) and oxygen (3.4),

serves as a mediator via electrostatic attraction. Subsequently, a silicon-based Lewis acid has been applied to achieve a C-SuFEx linkage reaction via the interactions of Si···F (BDE = 135 kcal/mol) with potassium organotrifluoroborates via propagation steps involving tricoordinate/tetracoordinate boron species. Herein, we report the application of C-SuFEx to link sulfonimidoyl fluorides and aryl/alkyl organotrifluoroborates via the electrostatic attraction of potassium cation with F and O prompting silicon-based Lewis acid to activate S(VI)-F bonds (Fig. 1C). The results of DFT theoretical calculations demonstrated that a lower Si-X (X = OTf, OMs, OAc, and Br) heterojunction bond energy for silicon-based Lewis acids and weak binding between $RBF_2$ and KX are the keys to the success of the propagation steps.

## Results

We commenced the C-SuFEx linkage reaction of sulfonimidoyl fluoride **1a** and phenyltrifluoroborate **2a** with the assistance of an initiator, and disappointingly, the metal-based Lewis acids failed to generate the corresponding product (Table 1, entry 1). Silicon-based Lewis acids were then tested due to the strong bonding energy between fluorine and silicon; fortunately, the desired sulfoximine product **3a** was detected in less than 10% yield in the presence of

[1]Shanghai Key Laboratory of Green Chemistry and Chemical Processes, School of Chemistry and Molecular Engineering, East China Normal University, 3663 North Zhongshan Road, Shanghai 200062, China. [2]State Key Laboratory of Petroleum Molecular and Process engineering, SKLPMPE, Sinopec research institute of petroleum processing Co., LTD., Beijing 100083, China; East China Normal University, Shanghai 200062, China. [3]School of Chemistry and Chemical Engineering, Henan Normal University, Xinxiang, Henan 453007, China. [4]These authors contributed equally: Suqin Zhao, Daming Zeng. ✉e-mail: wangming@chem.ecnu.edu.cn; xfjiang@chem.ecnu.edu.cn

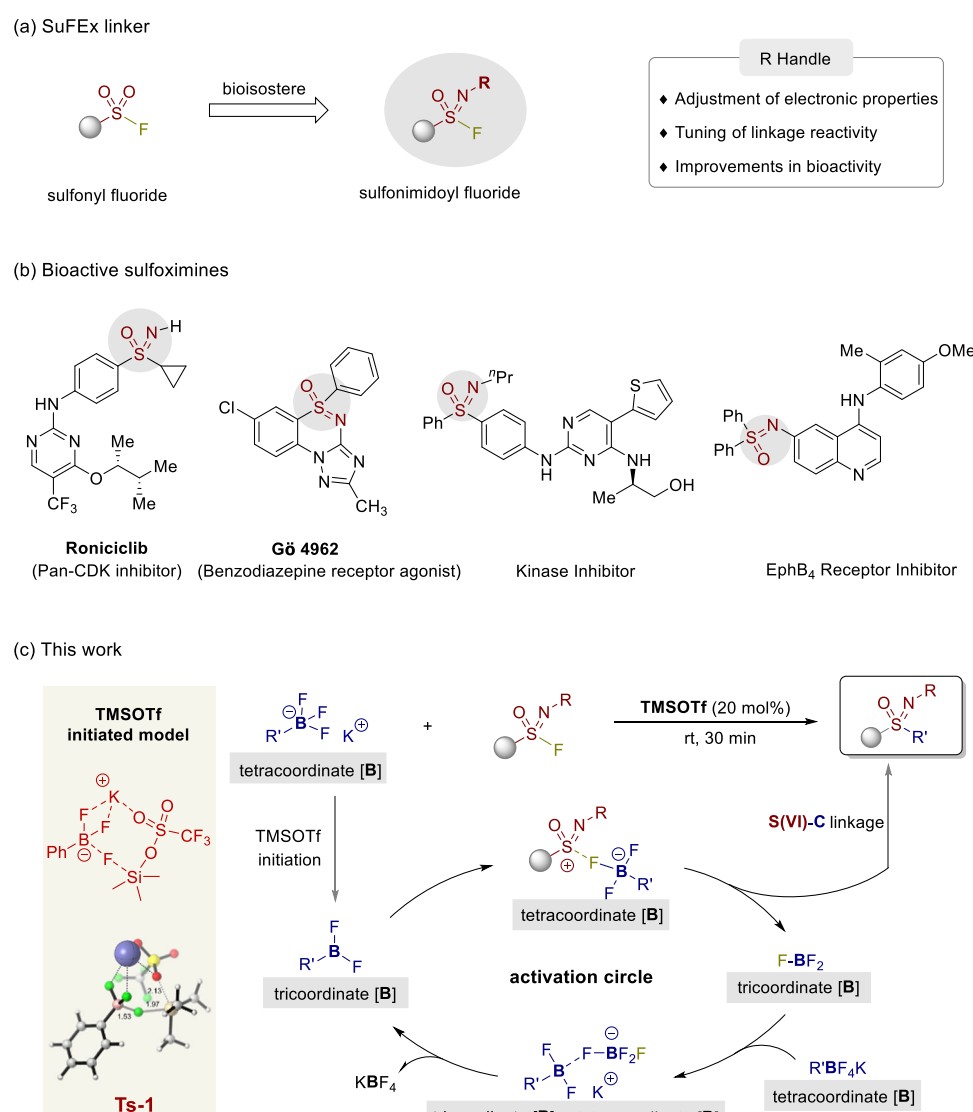

**Fig. 1 | SuFEx chemistry and sulfoximines. a** SuFEx linker, (**b**) bioactive sulfoximines, and (**c**) this work: the construction of diverse S(VI)-C bonds.

TMSCl, TMSBr and TMSOAc (entries 2–4). Unexpectedly, the substoichiometric amount of TMSOTf and TBSOTf delivered desired product **3a** in favorable yields (entries 5-6). However, a moderate yield was achieved by employing the Brønsted acid triflic acid as the initiator (entry 7). N-Tosyl and N-benzoxyl sulfonimidoyl fluorides failed to afford the corresponding C-SuFEx product **3**, and it is possible that electron-withdrawing groups are not conducive to stabilizing the sulfonimidoyl cation intermediate (entries 8-9). Further evaluation of the reaction solvents revealed that CH₃CN was the optimal solvent for the current C(Ar)-SuFEx linkage reaction (entries 10-11). When the ratio of **1a** to **2a** was tested, product **3a** was obtained in an almost quantitative yield with a 1.5:1 ratio (entry 12). Reducing the amount of TMSOTf at room temperature was not beneficial for the linkage (entries 13-14). However, raising the temperature slightly to 40 °C compensated for the impact of the lower quantity of Lewis acid on the C-SuFEx linkage efficiency, affording the sulfoximine in excellent yield (entry 15). It was found that cations of trifluoroborates other than potassium cation can still be compatible with the linkage (entries 16-18).

Subsequently, the linkage of sulfonimidoyl fluorides and organotrifluoroborates **2** was investigated comprehensively for a diverse sulfoximine library (Fig. 2). Substitution at the *ortho-*, *meta-*, and *para-* positions of the aromatic ring in potassium aryltrifluoroborate salts

with electron-poor or electron-rich functional groups allowed formation of the target sulfoximine products (**3b-3h**) in good yields. The presence of steric hindrance on the substrate barely affected the linkage efficiency (**3 h**). Conjugated aromatic ring systems, such as biphenyl, naphthalene, phenanthrene and pyrene, were well tolerated (**3i-3l**), especially for potassium (4-biphenyl)trifluoroborate, affording the product in a nearly quantitative yield. The linkage reaction was also efficient for potassium heteroaryltrifluoroborate salts with thienyl-, benzofuryl-, dibenzofuran-, and indolyl-derived substrates (**3m-3r**). The structure of **3 m** was further confirmed via X-ray diffraction analysis (CCDC: 2240513). Notably, this strategy enabled the conversion of alkenyl and allyl potassium trifluoroborate salts as SuFEx partners (**3 s** and **3t**). To further expand the library of functionalized sulfoximines, various types of sulfonimidoyl fluorides were evaluated. In addition to *ortho-*, *meta-*, and *para-*substituted aryl sulfonimidoyl fluorides (**3u-3aa**), substrates containing conjugated aromatic rings (**3ab-3ad**) and heterocycles (**3ae, 3af** and **3ag**) proved to be competent linkage partners. The linkage of N-Et-, N-iPr-, N-tBu-, N-Bn- and N-Ph-substituted sulfonimidoyl fluorides proceeded smoothly, affording the desired sulfoximines (**3ah-3al**). To our delight, alkyl and alkenyl sulfonimidoyl fluorides participated in the current C-SuFEx linkage with potassium trifluoroborate salts to generate the products in excellent yields (**3am-3ao**).

**Table 1 | Conditions optimization[a]**

| Entry | Initiator (equiv.) | M | R' | Solvent | Yield [%] |
|---|---|---|---|---|---|
| 1 | AlCl$_3$ or MgCl$_2$ (50 mol%) | K | n-Pr | CH$_3$CN | NP |
| 2 | TMSCl (50 mol%) | K | n-Pr | CH$_3$CN | <10 |
| 3 | TMSBr (50 mol%) | K | n-Pr | CH$_3$CN | NP |
| 4 | TMSOAc (50 mol%) | K | n-Pr | CH$_3$CN | NP |
| 5 | TMSOTf (50 mol%) | K | n-Pr | CH$_3$CN | 82 |
| 6 | TBSOTf (50 mol%) | K | n-Pr | CH$_3$CN | 82 |
| 7 | TfOH (50 mol%) | K | n-Pr | CH$_3$CN | 53 |
| 8 | TMSOTf (50 mol%) | K | Ts | CH$_3$CN | NP |
| 9 | TMSOTf (50 mol%) | K | Bz | CH$_3$CN | NP |
| 10 | TMSOTf (50 mol%) | K | n-Pr | DCM | 50 |
| 11 | TMSOTf (50 mol%) | K | n-Pr | THF | NP |
| 12[b] | TMSOTf (50 mol%) | K | n-Pr | CH$_3$CN | >99 |
| 13[b] | TMSOTf (20 mol%) | K | n-Pr | CH$_3$CN | 80 |
| 14[b] | TMSOTf (10 mol%) | K | n-Pr | CH$_3$CN | 11 |
| **15[b, c]** | **TMSOTf (20 mol%)** | K | **n-Pr** | **CH$_3$CN** | **97 (95)[d]** |
| 16[b, c] | TMSOTf (20 mol%) | Li | n-Pr | CH$_3$CN | 98 |
| 17[b, c] | TMSOTf (20 mol%) | Na | n-Pr | CH$_3$CN | 97 |
| 18[b, c] | TMSOTf (20 mol%) | Cs | n-Pr | CH$_3$CN | 98 |

[a]Reaction conditions: **1** (0.4 mmol), **2** (0.6 mmol), solvent (2 mL), N$_2$, 0 °C to rt, 30 min, NMR yields.
[b]**1** (0.6 mmol), **2** (0.4 mmol).
[c]rt to 40 °C.
[d]Isolated yields.

To further highlight the applicability of the current C-SuFEx linkage reaction, naturally occurring products and pharmaceuticals with sensitive functional groups and multiple heteroatoms were investigated (Fig. 3). The nonsteroidal anti-inflammatory drug valdecoxib allowed linkage with a benzothiophene and a naturally occurring estrone in good to excellent yields (**3ap** and **3aq**). The naturally occurring products coumarin and estrone could be linked efficiently via the current strategy (**3ar**). The C-SuFEx linkage of the highly selective COX-2 inhibitor celecoxib and the hypolipidemic drug clofibrate was readily achieved using this methodology (**3as**). Diacetone-fructose containing multiple heteroatoms was amenable to the TMSOTf-catalyzed linkage as well (**3at**). Additionally, the successful assembly of the anti-inflammatory drug flurbiprofen and the analgesic drug carprofen demonstrated the application value and prospects of this method for drug molecule linkage (**3au**).

To identify the mechanism of the TMSOTf-initiated linkage, the competition experiments of substituted sulfonimidoyl fluorides with different electron effects were conducted (Fig. 4). 4-OMe-substituted sulfonimidoyl fluoride afforded dramatically a higher yield compared to 4-CF$_3$-substituted sulfonimidoyl fluoride for the corresponding products, demonstrating that electron-rich sulfonimidoyl fluorides display a faster rate than the electron-poor sulfonimidoyl fluorides. Subsequently, $^{19}$F NMR studies were conducted (Fig. 5a). When 20 mol % TMSOTf was added to **1a** and **2a**, TMSF was generated in the reaction systems, demonstrating the activation from TMSOTf on both substrates **1a** and **2a**. Moreover, the chemical shift of **1a** shifted toward the high field region (Fig. 5a, I vs. VI), and new peaks appeared in the spectrum of **2a** (−152.04 and −152.09 ppm) with a peak area ratio of 1:4 (Fig. 5a, VII). When one equivalent of TMSOTf was added to substrate

**2a**, the peak attributed to **2a** (−143.59 ppm) and the abovementioned new peaks (−152.04 and −152.09 ppm) disappeared (Fig. 5a, VIII), but a peak attributed to PhBF$_2$ (−120.83 ppm) was observed. These results suggest that the new peaks may be attributed to a boron fluoride compound formed by PhBF$_2$ and unreacted **2a**. During the model reaction, the fluorine signal of **1a** did not change, indicating that TMSOTf interacted with **2a** in the linkage reaction first. At the same time, a signal attributed to KBF$_4$ was observed in the spectrum (Fig. 5a, IX).

Based on DFT theoretical calculations and verification (Fig. 5b), an energy barrier of 10.2 kcal/mol is required to obtain **Int-a** when TMSOTf is applied for the activation of substrate **1a**, and then the intermediate **Int-b** is obtained via the transition state **TS-a**. When TMSOTf interacts with substrate **2a**, the energy decreases to allow formation of **Int-1**, generating **Int-2** from **TS-1**, which is associated with an energy barrier of −24.7 kcal/mol. These results demonstrate that TMSOTf interacts with substrate **2a** preferentially in the reaction. During the activation process, the B-F bond length of **2a** is elongated from 1.44 Å (**Int-1**) to 1.53 Å (**TS-1**), resulting in the release of PhBF$_2$, TMSF, and KOTf. Subsequently, PhBF$_2$ activates the S-F bond in **1a**. The S(VI)-F bond length is extended from 1.67 Å (**Int-3**) to 1.85 Å (**TS-2**), resulting in the rapid release of the sulfonimidoyl cation (**Int-4**) by overcoming an energy barrier of 2.3 kcal/mol. Target product **3a** was obtained via phenyl migration, while BF$_3$ was dissociated with an energy barrier of −29.1 kcal/mol. During the activation process of BF$_3$, BF$_3$ may activate either substrate **1a** or **2a**, and the DFT calculations show that BF$_3$ tends to activate phenylboron trifluoride potassium salt **2a** (with an energy barrier of −6.8 kcal/mol) to afford **Int-5**, while the interaction between sulfonimidoyl fluorides and BF$_3$ is associated with

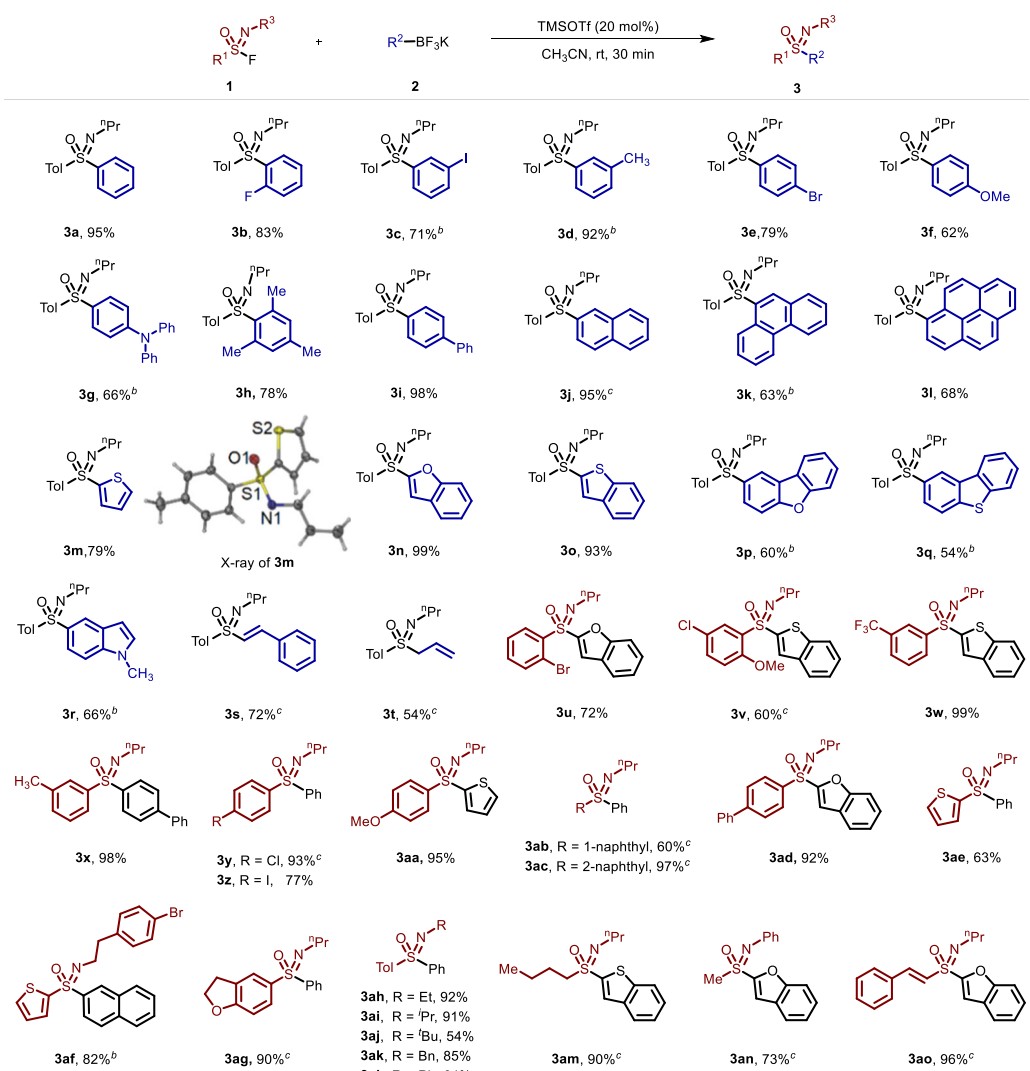

**Fig. 2 | Substrate scope[a].** The C-SuFEx linkage reaction of sulfonimidoyl fluorides and potassium trifluoroborate salts. [a]Reaction conditions: **1a** (0.6 mmol), **2a** (0.4 mmol), TMSOTf (20 mol%), CH$_3$CN (2 mL), N$_2$, rt to 40 °C, 30 min; [b]TMSOTf (50 mol%), rt, air, 30 min; [c]TMSOTf (30 mol%).

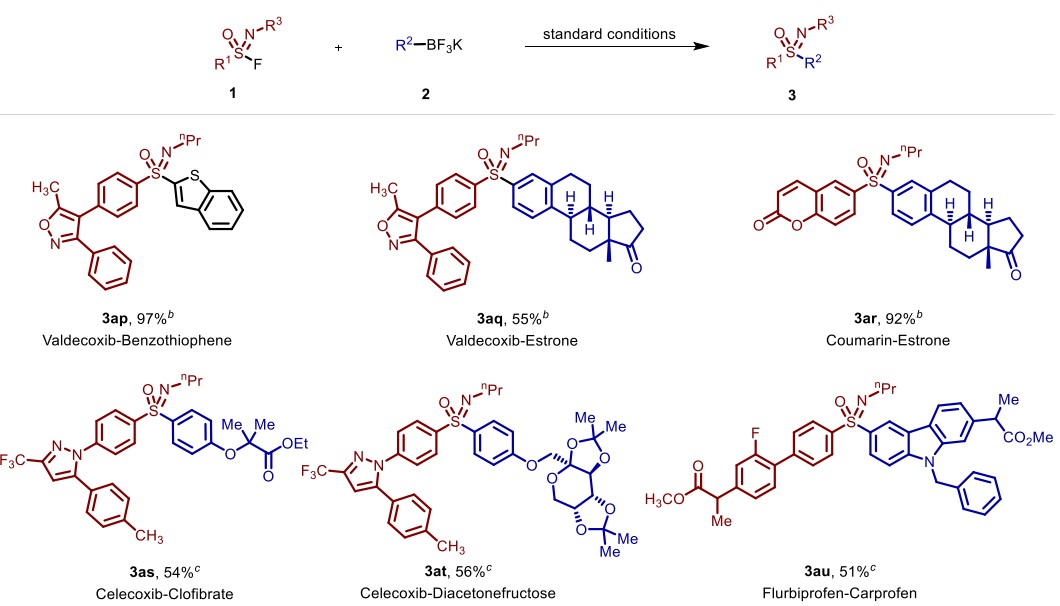

**Fig. 3 | The linkage of functional molecules[a].** [a]Reaction conditions: **1a** (0.6 mmol), **2a** (0.4 mmol), TMSOTf (20 mol%), CH$_3$CN (2 mL), N$_2$, rt to 40 °C, 30 min. [b]TMSOTf (30 mol%). [c]TMSOTf (50 mol%), rt, air.

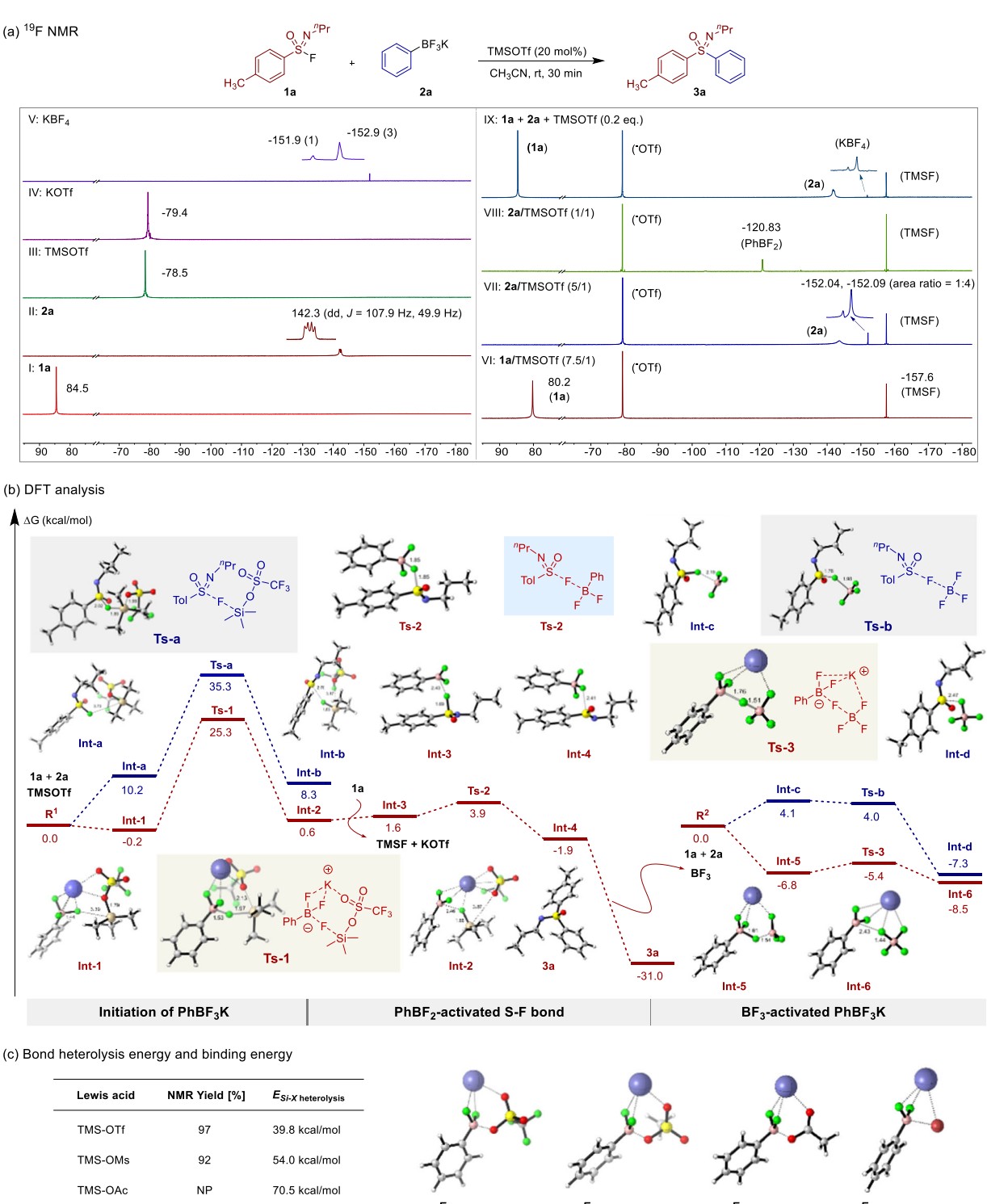

**Fig. 4 | Competition experiments.** Substituted sulfonimidoyl fluorides with different electron effects.

**Fig. 5 | Mechanistic studies. a** $^{19}F$ NMR studies in $CD_3CN$. **b** Density functional theory (DFT) calculation: the energy profiles were calculated at 298.15 K at the level of M06-2X/6-311 G(d,p) SMD=acetonitrile. **c** Bond heterolysis energies of Si-X (X = O, Br) and binding energies of $PhBF_2$ with KX.

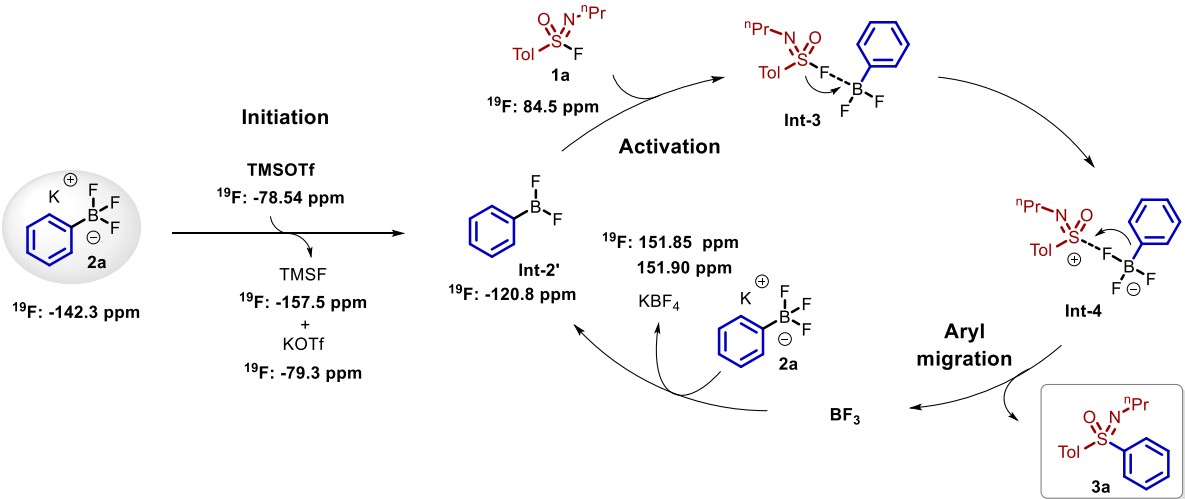

**Fig. 6 | The key catalytic cycle.** The catalytic cycle that including initiation, activation and aryl migration processes.

an energy barrier of 4.1 kcal/mol. Removal of the fluoride anion of **2a** was achieved under the activation of $BF_3$ to form **Int-6**, releasing $KBF_4$ and highly active $PhBF_2$, thus initiating the next activation cycle.

To further demonstrate the uniqueness of TMSOTf, different types of silicon-based Lewis acids were selected for control experiments (Fig. 5c). The yield of this linkage reaction was optimal with TMSOTf, and a slight decrease in the yield was observed when TMSOMs was used. No target product **3a** was generated with TMSOAc and TMSBr. A calculation of the reaction energy barriers showed that TMSOTf possessed a lower Si-O bond heterojunction energy and a lower energy barrier for different Lewis acids with potassium phenyltrifluoroborate **2a** (Supplementary Fig. 9). The potassium salt, generated by the reaction of the Lewis acid with **2a**, was not easily dissociated from $PhBF_2$, thereby hindering the reaction. The binding energies of different potassium salts and $PhBF_2$ were calculated, and the binding energy of KOAc was the highest (25.4 kcal/mol). Although the Si-Br bond heterolysis energy of TMSBr was moderate (50.4 kcal/mol vs. 70.5 kcal/mol), the energy barrier of the reaction of TMSBr with potassium phenyltrifluoroborate **2a** was higher (33.1 kcal/mol *vs.* 25.3 kcal/mol).

Based on these experimental results, a reaction mechanism was predicted, as shown in Fig. 6. Potassium phenyltrifluoroborate **2a** is first activated by TMSOTf to generate $PhBF_2$, releasing TMSF and KOTf. The S(VI)-F bond of sulfonimide fluoride **1a** is activated by $PhBF_2$ to obtain sulfonimide thiocation pair **Int-4** via **Int-3**. The aryl group migration afforded target product **3a**, releasing $BF_3$, which is activated with **2a** to regenerate phenylboron difluoride and $KBF_4$ to complete the activation process.

## Discussion

In conclusion, a TMSOTf-initiated C-SuFEx reaction of sulfonimidoyl fluorides and potassium organotrifluoroborates was established to synthesize a range of sulfoximines. A substoichiometric amount of TMSOTf activated stable and commercially available potassium organotrifluoroborates to release organodifluoroboranes, which interacted with sulfonimidoyl fluorides to generate $BF_3$ in situ as an activating reagent to achieve efficient linkage. A range of naturally occurring molecules and pharmaceuticals with multiple heteroatoms and sensitive functional groups were highly compatible with this linkage reaction, furnishing functional sulfoximines. Mechanistic studies and DFT calculations further revealed that trimethylsilyl triflate activated potassium organotrifluoroborates first and released organodifluoroboranes in situ. A lower Si-X (X = OTf, OMs, OAc, and Br) bond heterojunction energy of silicon-based Lewis acid and a weak binding

of $RBF_2$ with KX are the keys to the success of this C-SuFEx process. Further application of C-SuFEx of sulfonimidoyl fluorides is in progress in our laboratory.

## Methods

### General reaction procedure

Under nitrogen atmosphere, dry $CH_3CN$ (2 mL, 0.2 M) was added to a 10 mL oven-dried Schlenk tube, which was equipped with sulfonimidoyl fluoride **1** (0.60 mmol), potassium organotrifluoroborate salt **2** (0.40 mmol) and a stirring bar. TMSOTf (0.08 mmol) was dropwise added via the microsyringe at room temperature. Then the reaction mixture was stirred at 40 °C for 30 min and quenched with saturated aqueous solution of sodium carbonate. After the extraction of DCM for three times, the organic phase was washed with brine and dried over anhydrous $Na_2SO_4$. The solvent was evaporated under reduced pressure, and the residue was purified by column chromatography to yield the corresponding sulfoximine product **3**.

## Data availability

The X-ray crystallographic coordinates for the structures reported in this study have been deposited in the Cambridge Crystallographic Data Center (CCDC), under deposition number CCDC 2240513 (**3 m**). These data can be obtained free of charge from the Cambridge Crystallographic Data Center via www.ccdc.cam.ac.uk/data_request/cif. Experimental procedures, methods, characterization data, DFT calculations and NMR spectra are available in the Supplementary Information files and are also available from the corresponding author upon request. Source data are present in this paper.

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

## Acknowledgements

We are grateful for financial support provided by NSFC (22371075 (M.W.), 22125103 (X.J.) and 22071057 (M.W.)), Top-Notch Young Talents Program of China (202312A797 (M.W.)), STCSM (22JC1401000 (X.J.)), and the Fundamental Research Funds for the Central Universities.

## Author contributions

X.J. and M.W. conceived the idea and supervised the whole project. S.Z. and D.Z. contributed equally to this work. S.Z. and D.Z. carried out the experiments, and conducted the DFT calculations. X.J. and M.W. discussed the results, contributed to the writing of the manuscript, and commented on the manuscript. All authors approved the final version of the manuscript for submission.

## Competing interests

The authors declare no competing interests.

## Additional information

**Peer review information** : *Nature Communications* thanks the anonymous reviewers for their contribution to the peer review of this work. A peer review file is available.

