## [Peer Review File · Nature Communications]

C-SuFEx Linkage of Sulfonimidoyl Fluorides and OrganotrifluoroboratesReviewers' Comments:

Reviewer #1:

Remarks to the Author:

This manuscript by Wang, Jiang and co-workers constitutes a significant advance in the chemistry of sulfonimidoyl fluorides by successfully demonstrating their reaction with potassium trifluoroborates derivatives to prepare sulfoximines.

The reaction and the products obtained is likely to be of considerable value. Moving to the trifluoroborate derivatives will allow improved functional group compatibility and application compared to previous works. This is well demonstrated. There are a large number of examples and complex derivatives successfully prepared.

An important correction is the reference to a catalytic process - the discussion refers to catalytic species and a catalytic cycle. However, there is not a catalytic process in operation - there is not a catalytic species! The BF₃ generated provides a activating agent, and the B auto of BF₃ each time leaves the cycle as BF₄⁻ to activate the ArBF₃K species. It would be better described as propagation steps - this is not catalytic. The descriptions of the mechanism and relevant discussion should be updated accordingly throughout, including in the schemes.

The sulfonimidoyl fluoride reagent is used in excess. It would be nice to see some more examples of successful sulfonimidoyl fluoride reagents with different N-R groups. Bn and Ph are shown in Table 2. What is the result with a secondary alkyl, or Me derivatives for example. Is there a trend in reactivity for different Ar groups - eg PMP or 4-CF₃Ph, that could provide insight to mechanism/reactivity. Were electron poor heteroarene-BF₃ derivatives successful? eg pyridine derivatives, which would be particularly valuable.

The SI characterises the compounds suitably. There is relatively little further discussion in the SI. Further details on the preparation of the sulfonimidoyl fluoride reagents would be valuable to include.

There are several other changes in the phrasing and technical points that should be address, as listed below.

With these points duly addressed, the value in the reaction means this could be be suitable for publication in Nature Comm in the opinion of this referee.

Other changes and considerations

abstract -

delete - 'thus surpassing the achievements of previous studies'. The current work bring new insights, but this statement is not appropriate - all science being built upon previous studies.

rephrase BF₃ catalyst - see comments above.

In the first line of the introduction the authors refer to SuFex being 'first developed by Sharpless' The 2014 paper by Sharpless developed the term SuFex and extensively developed and and highlight the potential of this, but was not the first example of the reactivity. The authors should consider this and perhaps rephrase in the description of the history.

pg 2 - The references 19-22 appear out of place. these are not related to the C-SuFEx statement. change 'Subsequently, a silicon-based...' to 'Here, a silicon-based...' - assuming that this is referring to the current work. Otherwise a reference is missing to make the previous work clear.

Delete "in line with our research in organosulfur chemistry", and remove the corresponding references.

This appears unrelated to the current work, and should be removed.

the statement 'via the activation of K...F...Si' is unclear.

pg 3

Fig 1a - the boxed test in Fig 1A should be reconsidered -

- what is meant by electrical properties.

- there is no evidence of bioactivity improvement for the sulfinimidoyl fluorides vs sulfonyl fluorides themselves. The sulfoximine products is different.

rephrase 'the catalytic amount' with 'a substoichiometric amount'

pg 4

The discussion of the result with Ts and Bz appears to be conjecture. It refers to a cation intermediate - but this is not obvious from the calculations, nor other aspects of discussion. This should be rephrased.

what did happen with these derivatives in the Table 1?

pg 5 - 'cations other than phenyltrifluoroborate' - should this read potassium?

rephrase - did not affect the linkage.

pg 6 - delete 'extensively'.

Reviewer #2:

Remarks to the Author:

Jiang et al. reported the catalytic C-SuFEx linkage of sulfinimidoyl fluorides and potassium organotrifluoroborates. It is an imperative progress since the previous C-SuFEx linkage demands stoichiometric reagent. S(VI)-C(Ar) bonds are constructed promptly in minutes, and S(VI)-C(alkyl)/S(VI)-C(alkenyl) bonds can be realized via the current strategy as well. The ¹⁹F NMR studies and DFT calculations further revealed that the in situ formed BF₃ as the catalyst. A lower Si-X heterojunction bond energy for silicon-based Lewis acids and weak binding between RBF₂ and KX are the keys to the success of this catalytic cycle. This manuscript presents important contribution to SuFEx linkage and is a significant and meaningful discovery in the transformation and mechanism. It is suitable for publication in Nature Communications after minor revisions to the following issues.

1) The dosage of TMSOTf was described as 0.2 equiv. in Table1, but was described as 20 mmol% in Table 2. Keep the description consistent.

2) This reaction is conducted under N₂ conditions. What is the result under air atmosphere?

3) "3ao" should be corrected to "3aq" in the description of drug molecular link.

4) "R'" should be corrected to "nPr" in Figure 2a.

5) "1a" and "2a" should be corrected to "1" and "2" in Table 1.

Reviewer #3:

Remarks to the Author:

In this paper, Jiang and co-workers reported a C-SuFEx catalytic reaction between sulfinimidoyl fluorides and aryl/alkyl organotrifluoroborates, with BF₃ as a catalyst. Although the authors wrote the S(VI)-C bond formation of sulfinimidoyl fluorides is limited to coupling with TMSCF₃ and highly active organometallic reagents such as Grignard and lithium reagents. Previously, several papers (Organic Letters (2023), 25(30), 5591-5596; Angewandte Chemie, International Edition (2022), 61(44), e202207100, etc.) have been reported for similar constructions of C-S bonds. Moreover, this conversion released BF₃, a highly toxic compound, as a catalyst and generated harmful fluoroborides

in the reaction, which is contrary to green chemistry. Publication of this to Nature Communications is not recommended.

Some issues need to be addressed:

- 1) In the Introduction part, "However, the C-SuFEx [SuFEx for S(VI)-C bond formation] of sulfonimidoyl fluorides is limited to coupling with TMSCF₃ and highly active organometallic reagents such as Grignard and lithium reagents", but several papers (Organic Letters (2023), 25(30), 5591-5596; Angewandte Chemie, International Edition (2022), 61(44), e202207100) don't look like what the author described.
- 2) In the Results part, the authors described "The presence of steric hindrance on the substrate barely affected the linkage efficiency." However, the yields of compounds 3k and 3l were significantly lower.
- 3) Most compounds have HRMS data that is either identical to the calculated HRMS or within +0.0001-0.0002 higher than the calculated HRMS. But most of the HRMS data reported by the authors exceeded this range in the SI.

Responds to the reviewers' comments

Reviewer #1: Minor Revisions

Question 1: An important correction is the reference to a catalytic process - the discussion refers to catalytic species and a catalytic cycle. However, there is not a catalytic process in operation - there is not a catalytic species! The BF₃ generated provides a activating agent, and the B auto of BF₃ each time leaves the cycle as BF₄- to activate the ArBF₃K species. It would be better described as propagation steps - this is not catalytic. The descriptions of the mechanism and relevant discussion should be updated accordingly throughout, including in the schemes.

Answer 1: Thanks for the referee's reminding and a great suggestion. Indeed, this is not a catalytic process, though TMSOTf is a catalytic amount of activating reagent. According to the referee's suggestions, the descriptions of "catalytic cycle" and "catalytic species" have been corrected, and it has been described as "propagation steps" and "activating reagent". The revisions were shown in "track changes" form.

Question 2: It would be nice to see some more examples of successful sulfonimidoyl fluoride reagents with different N-R groups. Bn and Ph are shown in Table 2. What is the result with a secondary alkyl, or Me derivatives for example. Is there a trend in reactivity for different Ar groups - eg PMP or 4-CF₃Ph, that could provide insight to mechanism/reactivity.

Answer 2: More examples of sulfonimidoyl fluorides with different *N*-R groups have been added in the revised manuscript. Primary alkyl (-Et), secondary alkyl (-ⁱPr), and tertiary alkyl (-^tBu) -derived sulfonimidoyl fluorides proceeded smoothly, affording the desired sulfoximine products **3**.

The competition experiments of substituted sulfonyl fluorides with different electron effects were conducted. 4-OMe-substituted sulfonyl fluoride afforded a dramatically higher yield compared to 4-CF₃-substituted sulfonyl fluoride for the corresponding products, demonstrating that electron-rich sulfonyl fluorides display a faster reaction rate than the electron-poor sulfonyl fluorides.

Question 3: Were electron poor heteroarene-BF₃ derivatives successful? eg pyridine derivatives, which would be particularly valuable.

Answer 3: Thanks for the referee's suggestion. Since basic pyridine formed a salt with the activating reagent TMSOTf, pyridine-derived trifluoroborate (-BF₃K) is not good candidate for the coupling partner. Luckily, other heteroaromatic trifluoroborates are well compatible with the current linkage, such as thienyl-, benzofuryl-, dibenzofuran-, and indolyl-derived trifluoroborates.

Question 4: The SI characterises the compounds suitably. There is relatively little further discussion in the SI. Further details on the preparation of the sulfonimidoyl fluoride reagents would be valuable to include.

Answer 4: Thanks for the referee's reminding. The general procedure for the synthesis of sulfonimidoyl fluorides has been added in the Section II of supporting information (Page S3-S5).

II. General procedure for the synthesis of sulfonimidoyl fluorides.

Synthesis of sulfinamides:

Method A:

(COCl)₂ (11.0 mol, 1.1 equiv.) was added dropwise into a 50 mL flame-dried Schlenk flask at 0 °C, which was equipped with sodium sulfinate **a** (10.0 mmol, 1.0 equiv.) and anhydrous toluene (25.0 mL, 0.4 M). After dropping, the reaction solution was

transferred to room temperature for 1 h to form the sulfinyl chloride. Then the reaction solution was added dropwise to the toluene (25.0 mL, 0.4 M) solution of triethylamine (15.0 mmol, 1.5 equiv.) and amine (13.0 mmol, 1.3 equiv.) in another 100 mL Schlenk flask at 0 °C. The mixture was stirred for 1 h at room temperature and quenched with water. After the extraction of ethyl acetate for three times, the organic phase was washed with brine and dried over anhydrous Na₂SO₄. The solvent was evaporated under reduced pressure, and the residue was purified by column chromatography to yield the corresponding sulfinamide **d**.

Method B:

Under nitrogen atmosphere, a DCM (25.0 mL, 0.4 M) solution of amine (10.0 mmol, 1.0 equiv.), triphenylphosphine (10.0 mmol, 1.0 equiv.) and triethylamine (20.0 mmol, 2.0 equiv.) was added dropwise via syringe pump (8 mL/h) to a solution of sulfonyl chloride **b** (10.0 mmol, 1.0 equiv.) in anhydrous DCM (25.0 mL, 0.4 M) at 0 °C. The mixture was stirred for 18 h at 0 °C. After TLC monitoring reaction, the solvent was evaporated under reduced pressure, and the residue was purified by column chromatography to yield the corresponding sulfinamide **d**.

Method C:

SO₂Cl₂ (15.5 mmol, 3.1 equiv.) was added dropwise into a dry 25 mL oven-dried Schlenk tube, which was equipped with dimethyl disulfide **c** (5 mmol, 1 equiv.) and acetic acid (10 mmol, 2 equiv.) at -20 °C. The mixture was then stirred at -20 °C for 3 h and then warmed to rt for 2 h. Subsequently, the reaction mixture was transferred to 35 °C for 1 h. After completion, the volatiles were evaporated under reduced pressure and the resulting methylsulfinyl chloride was directly used for the next step without purification. The DCM (15.0 mL, 0.67 M) solution of methylsulfinyl chloride was slowly added into a solution of aniline (10.0 mmol, 2 equiv.) in dry DCM (10 mL, 1 M). Then the mixture was warmed to room temperature for 3 h and quenched with water. After the extraction of DCM for three times, the organic phase was washed with brine and dried over anhydrous Na₂SO₄. The solvent was evaporated under reduced pressure, and the residue was purified by column chromatography to yield the corresponding *N*-phenylmethane sulfinamide.

Synthesis of sulfonimidoyl fluorides:

NCS (5.5 mmol, 1.1 equiv.) was added portionwise into a dry 100 mL round-bottom flask, which was equipped with sulfonamide **d**, TBAF (5.5 mL, 1 M in THF, 1.1 equiv.), CH₃CN (40 mL, 0.125 M) and a stirring bar at 0 °C. The mixture was warmed to room temperature for 30 min. After TLC monitoring reaction, the solvent was evaporated under reduced pressure, and the residue was purified by column chromatography to yield the corresponding sulfonimidoyl fluorides **1**.

Question 5: delete - ‘thus surpassing the achievements of previous studies’. The current work bring new insights, but this statement is not appropriate - all science being built upon previous studies. -rephrase BF₃ catalyst - see comments above.

Answer 5: Thanks for the referee’s reminding. The description of “thus surpassing the achievements of previous studies” has been deleted, and “BF₃ catalyst” has been corrected to “BF₃ as an activating reagent”.

Question 6: In the first line of the introduction the authors refer to SuFex being ‘first developed by Sharpless’ The 2014 paper by Sharpless developed the term SuFex and extensively developed and highlight the potential of this, but was not the first example of the reactivity. The authors should consider this and perhaps rephrase in the description of the history.

Answer 6: Thanks for the referee’s reminding. We want to highlight the concept of SuFEx was introduced by Sharpless and coworkers in the first sentence. According to the referee’s suggestion, this sentence has been rephrased as “Since the concept of sulfur(VI) fluoride exchange (SuFEx) was first introduced by Sharpless and coworkers in 2014”.

Question 7: pg 2 - The references 19-22 appear out of place. these are not related to the C-SuFEx statement.

Answer 7: Thanks for the referee's reminding. The references 19-22 should be removed to the previous sentence, which describes the discovery of sulfoximide-containing drugs. The typos have been corrected in this revised manuscript.

Question 8: change 'Subsequently, a silicon-based...' to 'Here, a silicon-based...' - assuming that this is referring to the current work. Otherwise a reference is missing to make the previous work clear.

Answer 8: Thanks for the referee's reminding. This sentence is to describe our current strategy. According to this referee's suggestion, "Subsequently, a silicon-based..." has been rephrased as "Herein, a silicon-based..."

Question 9: Delete "in line with our research in organosulfur chemistry", and remove the corresponding references. This appears unrelated to the current work, and should be removed.

Answer 9: Thanks for the referee's suggestion. The sentence of "in line with our research in organosulfur chemistry" has been deleted and the corresponding references have been removed.

Question 10: the statement 'via the activation of K...F...Si' is unclear.

Answer 10: The results of DFT theoretical calculations revealed the interactions of potassium cation with F (from trifluoroborates) and O (from TMSOTf) in the activation model of TMSOTf with trifluoroborates.

This description has been rewritten as "Herein, we report the application of C-SuFEx to link sulfonimidoyl fluorides and aryl/alkyl organotrifluoroborates via the electrostatic attraction of potassium cation with F and O, prompting silicon-based Lewis acid to activate S(VI)-F bonds" for clearer expression.

Question 11: pg 3

Fig 1a - the boxed text in Fig 1A should be reconsidered –

- what is meant by electrical properties.
- there is no evidence of bioactivity improvement for the sulfinimidoyl fluorides vs sulfonyl fluorides themselves. The sulfoximine products is different.
- rephrase ‘the catalytic amount’ with ‘a substoichiometric amount’

Answer 11: 1) The description “electrical properties” mean the R handle can be used to regulate the electrical properties of sulfonyl fluoride. For example, when R is an electron-withdrawing group, the corresponding sulfonyl fluoride displays higher electrophilicity.

2) Thanks for the referee’s reminding. The inaccurate description of “Improvement in bioactivity” has been corrected to “Regulating of bioactivity”.

3) The description of “the catalytic amount” has been corrected to “a substoichiometric amount” according to the referee’s great suggestion.

Question 12: pg 4

The discussion of the result with Ts and Bz appears to be conjecture. It refers to a cation intermediate - but this is not obvious from the calculations, nor other aspects of discussion. This should be rephrased.

-what did happen with these derivatives in the Table 1?

Answer 12: According to the DFT calculations, a sulfonimidoyl cation intermediate was proposed in the current transformation. Since this is not an absolute conclusion, this sentence has been rephrased as “it is possible that electron-withdrawing groups are not conducive to stabilizing the sulfonimidoyl cation intermediate” according to the referee’s suggestion.

When *N*-Tosyl and *N*-benzoxyl sulfonimidoyl fluorides were conducted in the current reaction, the corresponding sulfonimidoyl fluorides were decomposed without desired linkage product obtained.

Question 13: pg 5

‘cations other than phenyltrifluoroborate’ - should this read potassium?

rephrase - did not affect the linkage.

Answer 13: Yes, it means cations other than potassium cation. This sentence has been rephrased as “It was found that cations of trifluoroborates other than potassium cation can still be compatible with the linkage”.

Question 14: pg 6

- delete 'extensively'.

Answer 14: According to the referee’s suggestion, “extensively” has been deleted.

Reviewer #2: Minor Revisions

Question 1: The dosage of TMSOTf was described as 0.2 equiv. in Table1, but was described as 20 mmol% in Table 2. Keep the description consistent.

Answer 1: Thanks for the referee’s reminding. The amount of TMSOTf has been described as 20 mmol% in the entire manuscript to keep consistency.

Question 2: This reaction is conducted under N₂ conditions. What is the result under air atmosphere?

Answer 2: When the model reaction was conducted under air atmosphere, only 29% yield of **3a** was afforded. Under oxygen atmosphere, a yield of 93% can still be obtained. These results indicate that the water in the air gave the significant impact on the current linkage efficiency.

Question 3: “3ao” should be corrected to “3aq” in the description of drug molecular link.

Answer 3: Thanks for the referee’s reminding. The typo of “3ao” has been corrected to “3aq”.

Question 4: “R” should be corrected to “nPr” in Figure 2a.

Answer 4: Thanks for the referee’s reminding. The typo of “R” has been corrected to “nPr” in Figure 2a.

Question 5: “1a” and “2a” should be corrected to “1” and “2” in Table 1.

Answer 5: Thanks for the referee’s reminding. “1a” and “2a” has been renumbered as “1” and “2” in Table 1.

Reviewer #3:

Question 1: Although the authors wrote the S(VI)-C bond formation of sulfonimidoyl fluorides is limited to coupling with TMSCF_3 and highly active organometallic reagents such as Grignard and lithium reagents. Previously, several papers (*Organic Letters* (2023), 25(30), 5591-5596; *Angewandte Chemie, International Edition* (2022), 61(44), e202207100, etc.) have been reported for similar constructions of C-S bonds.

Answer 1: Thanks for the referee's question. The current work is totally different from the previous two works mentioned by the referee, and we made a comparison with two previous works as follows.

The comparison of our previous works (*Org. Lett.* **2023**, 25, 5591. *Angew. Chem. Int. Ed.* **2022**, 61, e202207100.) **and the current work:**

(I) Different reaction types

Angew's work: **2 equiv. of $\text{BF}_3 \cdot \text{OEt}_2$** catalyzed C-SuFEx of sulfonimidoyl fluorides and unactivated alkenes was established via hydrosulfonimidoylation to construct S(VI)-C_{alkyl} (sp^3) bonds.

Org Lett's work: **2 equiv. of $\text{BF}_3 \cdot \text{OEt}_2$** catalyzed C-SuFEx of sulfonimidoyl fluorides and allyltrimethylsilanes was developed to construct S(VI)-C_{allyl} (sp^3) bonds.

The current work: **A catalytic amount of TMSOTf** initiated C-SuFEx of sulfonimidoyl fluorides and aryl/alkyl organotrifluoroborates was achieved to construct S(VI)-C_{allyl} (sp^3), S(VI)-C_{alkenyl} (sp^2) and S(VI)-C_{aryl} (sp^2) bonds.

(II) Different reaction mechanisms

Angew's work: The sulfonimidoyl cationic intermediate underwent an intermolecular hydrogen atom transfer (HAT) via a concerted transient state to afford the products.

Org Lett's work: The linkage was initiated with the activation of sulfonimidoyl fluoride and followed by the transfer of fluoride anion to TMS group.

The current work: The linkage was initiated with the electrostatic attraction of potassium cation with F and O prompting silicon-based Lewis acid to activate S(VI)-F

bonds. A lower Si-O heterojunction bond energy for TMSOTf and weak binding between RBF₂ and KOTf are the keys to achieve the catalytic process.

Question 2: Moreover, this conversion released BF₃, a highly toxic compound, as a catalyst and generated harmful fluoroborides in the reaction, which is contrary to green chemistry.

Answer 2: Thanks for the referee's question. The current reaction did not generate BF₃ finally, but rather KBF₄, which is a stable and non-toxic salt at room temperature. The BF₃ generated in situ will continue to participate in the reaction as a catalyst, ultimately turning it into KBF₄ salt.

Question 3: In the Introduction part, "However, the C-SuFEx [SuFEx for S(VI)-C bond formation] of sulfonyl fluorides is limited to coupling with TMSCF₃ and highly active organometallic reagents such as Grignard and lithium reagents", but several papers (*Organic Letters* (2023), *25*(30), 5591-5596; *Angewandte Chemie*,

International Edition (2022), 61(44), e202207100) don't look like what the author described.

Answer 3: The C-SuFEx of sulfonimidoyl fluorides with TMSCF_3 is the works of reference 23-24. (*Org. Lett.* **2011**, *13*, 768. *Angew. Chem. Int. Ed.* **2019**, *58*, 4552.), and the C-SuFEx of sulfonimidoyl fluorides with Grignard and lithium reagents is the works of reference 17, 25 (*Angew. Chem. Int. Ed.* **2018**, *57*, 1939. *Chem. Commun.* **2022**, *58*, 5387.). Our continuous contribution is (*Org. Lett.* **2023**, *25*, 5591. and *Angew. Chem. Int. Ed.* **2022**, *61*, e202207100.), which are already cited.

Question 4: In the Results part, the authors described “The presence of steric hindrance on the substrate barely affected the linkage efficiency.” However, the yields of compounds **3k** and **3l** were significantly lower.

Answer 4: In the manuscript, our descriptions are “The presence of steric hindrance on the substrate barely affected the linkage efficiency (**3h**)” and “Conjugated aromatic ring systems, such as biphenyl, naphthalene, phenanthrene and pyrene, were well tolerated (**3i**, **3j**, **3k**, and **3l**)”. As the substrate with the highest steric hindrance, the yield of **3h** is good. The slight decrease in the yields of **3k** and **3l** is not the reason of steric hindrance, but due to the electron deficiency of conjugated aromatic groups.

3h, 78%

3k, 63%

3l, 68%

Question 5: Most compounds have HRMS data that is either identical to the calculated HRMS or within +0.0001-0.0002 higher than the calculated HRMS. But most of the HRMS datas reported by the authors exceeded this range in the SI.

Answer 5: Thanks for the referee's reminding. However, all our HRMS data are within ± 0.0009 in the SI, which are enough to prove the correctness of the elements in

these compounds. Our mass spectra were recorded on an Agilent Technologies 6224 TOF LC/MS, which was accepted by all the publication.

Reviewers' Comments:

Reviewer #1:

Remarks to the Author:

The authors have suitably addressed the comments from the referees. The removal of the term catalytic is appropriate. The additional examples of different types of N-alkyl derivatives are also a useful addition.

My opinion is that the nucleophile types are sufficiently different to those related reactions previously reported, that means this work is suitable for Nature Comm. I would however recommend that the OL 2023 reference cited by referee 3 (OL 2023, 5591) is added to the references.

My only remaining minor comments relate to Fig 1A. The description of 'R Handle' is not clear. Should electrical properties read electronic properties? I still don't understand the regulating of biological activity. Is this referring to the sulfonimidoyl fluoride as a covalent probe - in which case what is the role of the R group (vs the other C linked group) - or the sulfoximines derived from it?

Reviewer #2:

Remarks to the Author:

The authors have well addressed the questions raised the reviewers. I satisfy the revision. I support its publication in Nature Communications.